# Structural Safety Evaluation of Precast, Prestressed Concrete Deck Slabs Cast Using 120-MPa High-Performance Concrete with a Reinforced Joint

**DOI:** 10.3390/ma12183040

**Published:** 2019-09-19

**Authors:** Jae-Hyun Bae, Hoon-Hee Hwang, Sung-Yong Park

**Affiliations:** 1R&D Center, Korea Road Association, Seongnam-si 13647, Korea; bjh@kroad.or.kr; 2Department of Infrastructure Safety Research, Korea Institute of Civil Engineering and Building Technology, Goyang-si 10223, Korea; sypark@kict.re.kr

**Keywords:** ultra-high-performance concrete, deck slab, prestressed concrete, static test, fatigue test, deck slab joint

## Abstract

Prestressed concrete structures are used in various fields as they can reduce the cross-sectional area of members compared with reinforced concrete structures. In addition, the use of high-performance and strength concrete can help reduce weight and achieve excellent durability. Recently, structures have increasingly been constructed using high-performance and strength concrete, and therefore, structural verification is required. Thus, this study experimentally evaluated the structural performance of a long-span bridge deck slab joint, regarded as the weak point of structures. The specimens were designed with a 4 m span for application to cable-stayed bridges. To ensure the required load resistance and serviceability, the specimens comprised of 120 MPa high-performance fiber-reinforced concrete and were prestressed. The deck slabs satisfied all static and fatigue performance as well as serviceability requirements, although they were thinner than typical concrete bridge deck slabs. The study also verified whether the deck slabs were suitable to help implement precast segmental construction methods. Finally, the results confirmed that the structural performance of the developed prestressed concrete (PSC) deck slab was sufficient for the intended bridge application as it achieved a sufficiently large safety factor in the static and fatigue performance tests, relative to the design requirement.

## 1. Introduction

High-performance concrete not only improves the durability of structures by ensuring high quality, but also enables weight reduction owing to its excellent structural performance. Recently, high-performance materials such as ultra-high-performance concrete, whose strength has been significantly increased compared with normal concrete, have been attracting attention as they help realize innovative performance improvements and light-weight structures [1,2]. 

Bridge deck slabs are frequently damaged members because they directly support live loads through contact with the wheel load of vehicles. Therefore, continuous efforts have been invested to improve the quality and structural performance of these slabs by utilizing high-performance materials. In particular, many research and construction examples that utilize a precast method for accelerated bridge construction can be found in the literature [3]. The precast deck should be assembled with several prefabricated panels to complete the deck members; thus, there is a risk of lack of continuity. Therefore, joints between panels that can guarantee continuity are necessary; further, their safety must be verified as they could potentially become weak points of the structure. Several studies have been conducted on the joints of precast decks, and various types of joint details such as female–female joints, match–cast joints, male–to–female joints, spiral confinement methods, lap-spliced joints, and post-tensioning methods have been proposed [4]. All these joints have been experimentally verified to provide adequate structural performance; however, their details need to be improved considering the characteristics of high-performance materials. The bridge deck slab to which the high-performance materials are applied can exhibit high load-resistance performance, irrespective of their thickness and help reduce the weight of structures; therefore, joints with suitable joint details are required. The performance of ultra-high-performance concrete (UHPC) joints applied to the precast deck was evaluated using several joint details that could utilize the high adhesion performance of the UHPC [5]. Experiments on UHPC lap-spliced joints reported that the lap-spliced length of the steel rebar required to achieve the same level of flexural performance as without the joint was 200 mm [6]. The experimental research on the ultra-high-performance fiber reinforced concrete (UHPFRC) joint between precast concrete panels reinforced with glass-fiber-reinforced polymer (GFRP) bar concluded that using a lap-spliced length of 200 mm can maintain continuity of joints until failure occurs in the adjacent precast body [7]. Another study demonstrated that polymer methyl methacrylate-polymer concrete (PMMA-PC) could be an alternative to economic joint filler materials owing to their extremely high adhesion performance. In particular, the required lap-spliced length is smaller than UHPC and 40% of normal concrete [8].

These previous studies were focused on RC–type–joints because of the high adhesion performance of UHPC. However, research on prestressed concrete (PSC)–type–joints suitable for a long-span structure is relatively insufficient.

In this study, the behavior of joints in thin long-span bridge deck slabs was evaluated experimentally. The specimens were fabricated with a thickness of 150 mm and a span of 4 m. The high-performance concrete of 120 MPa was applied to resist the flexural moment caused by this relatively thin long span. A method to apply prestress using post-tension was adopted for this type of joint. This was because it could address problems such as deflection and cracks that could be disadvantageous owing to the relatively thin thickness of the slabs.

## 2. Description of Specimens

### 2.1. Specimens

To verify the structural performance of the 120 MPa prestressed concrete (PSC) deck slab with reinforced steel fibers, various specimens were designed and fabricated, as summarized in Table 1. To verify the performance of the deck slabs with joints, a specimen without a joint (SC120f-PSC-NJ) was fabricated as a reference. In addition, two specimens for static performance tests (SC120f-PSC-J1, J2) and a specimen for fatigue performance testing (SC120f-PSC-JF) were fabricated.

The cross-section of the deck slab specimens was designed in accordance with the Korean Highway Bridge Design Code (Limit State Design) (2015) [9] and the Structural Design Guideline for Fiber-Reinforced SUPER Concrete [10]. The dimensions of the specimens were 2 m (W) × 4.2 or 5.4 m (L) × 0.15 m (H). Those specimens with a length of 4.2 m were used for static testing. The fatigue test specimen was 5.4 m long, longer than the static test specimen, because additional space was required to prevent the specimen from being dislodged from its supporting points as a load was repeatedly applied. In the experiments, a relatively long span of 4.0 m was adopted, given that the deck slabs were primarily investigated for application to cable-stayed bridges. For those specimens with joints, the interfaces where the segments met were fitted with shear keys to improve the performance by mechanical interlocking. The types and cross-sectional joint details of the test specimens based on the purpose of the experiment are summarized in Table 1 and shown in Figure 1.

The tendon material was SWPC 7BL, which is a low-relaxation steel with seven strands and a 15.2 mm nominal diameter. The yield and tensile strengths of the strands were 1600 and 1881 MPa, respectively. Full prestressing was applied using the post-tension method to prevent tensile stress being applied to the overall section, while the tendons were placed in a straight line in groups of five at 1.0 m intervals. Moreover, the eccentricity of the tendons was 20 mm, and flat-type anchors, typically used for prestressed deck slabs, were used.

### 2.2. Material Properties of SC120f

High-strength concrete with a compressive strength of about 120 MPa, sustained tensile strength through internal fiber reinforcement, and exceptional durability as compared with conventional concretes was applied to the test specimen. The properties of the materials used in SC120f are listed in Table 2. Ordinary Portland cement was adopted as cement, and silica fume was used as reactive powder. Fine aggregates comprising quartz sand, with diameters smaller than 0.5 mm, and SiO_2_ content larger than 90%, were applied to SC120f. Coarse aggregates were not used. The steel fiber used exhibited tensile strength higher than 2000 MPa and a diameter of 0.2 mm. Further, polycarboxylate superplasticizer (density 1.01 g/cm^3^, dark brown liquid, solid content 30%) was used. A combination of calcium sulfoaluminate (CSA) type expansion admixture and glycol-based shrinkage reducing agent was used as the shrinkage reducing agent.

### 2.3. Fabrication

All the specimens, except for SC120f-PSC-NJ with no joint, were fabricated by match casting. After installing a formwork, the rebar and tendons were placed, and then the concrete was first poured into one side of the joint. The poured concrete was allowed to cure for approximately 2–3 days, and then the concrete was poured into the other side using match casting. Subsequently, high-temperature steam curing (at about 90 °C) was conducted. After curing, epoxy was applied to the interface of the joint where the segments abutted, and prestressing was applied using a hydraulic jack. At this time, the jacking force was determined to maintain the effective prestress considering the immediate losses such as elastic shortening of the concrete, anchorage losses, and frictional losses. After jacking, grouting was conducted to complete the specimens (Figure 2).

## 3. Static Performance Evaluation

### 3.1. Static Loading Test (SC120f-PSC-NJ, J1, and J2 Specimens)

In the static loading tests, the deflection, strain, and joint widening of each deck slab specimen were measured, and the behavior of the specimens was examined using a 3500-kN static actuator (MTS, Eden Prairie, MN, USA), with the load being applied until failure. The load was applied using a displacement-controlled method at a rate of 3 mm/min. Furthermore, the load was applied using a third-point loading method by ASTM C78/C78M [11] such that the largest bending moment would be generated at the center of the specimen (where the joint was located). Under actual boundary conditions, the bridge deck slab would be combined with the girder or crossbar; however, simple support was applied to induce conservative experimental results (Figure 3).

### 3.2. Result of Static Loading Test

#### 3.2.1. Failure Mode

The results of the static load test revealed that the SC120f-PSC-NJ specimen without the joint exhibited typical flexural behavior similar to general beam members. As the load increased, many tensile cracks appeared and grew at the bottom of the deck, while the concrete was crushed at the top of the deck slab between the load points during the final failure. At the moment of failure, the maximum load was 435.65 kN, and the deflection of the center of the deck slab under the maximum load was 88.06 mm.

For the SC120f-PSC-J1 and J2 specimens with joints, joint widening occurred from the start of loading, and then cracking occurred in the concrete adjacent to the joint. The joint widened as the load increase, and failure occurred as the concrete at the top of the joint was crushed. The SC120f-PSC-J1 and J2 specimens with joints had relatively few cracks compared with the SC120f-PSC-NJ specimen without a joint, given that the cracks form around the initial cracks initiated by the separation of the joints. The measured maximum loads were 322.07 and 289.04 kN for the SC120f-PSC-J1 and J2 specimens, respectively, while the deflections of the center of the deck slab under the maximum load were 58.6 and 60.18 mm, respectively (Figure 4).

#### 3.2.2. Evaluation of Flexural Strength

To evaluate the level of safety in terms of strength, the experimental results were compared with the ultimate limit state load required by the design code. The ultimate limit load was calculated by considering the third-point load test conditions from the ultimate moments induced by the ultimate limit state load combination specified in the design code. As a result of verifying the structural performance of the PSC deck slabs with joints, the maximum loads of the SC120f-PSC-J1 and J2 specimens were 74% and 66% of that of SC120f-PSC-NJ, respectively. Although the presence of the joints caused the maximum load to decrease, the values were 3.67 and 3.29 times higher than the ultimate limit load specified in the design code, as shown in Figure 5 and Figure 6. Therefore, the fabricated PSC deck slabs had a sufficient safety factor compared with the design requirement despite the presence of a joint (Table 3).

### 3.3. Evaluation of Serviceability

The Korean Highway Bridge Design Code (Limit State Design) specifies that deflection of a bridge concrete deck slab caused by live loads and their dynamic effects must not exceed the following limits, where L is the distance between the centers of the deck slab supports: deck slab with no passage of people, L/800; deck slab with passage of a limited number of people, L/1000; deck slab with passage of many people, L/1200. Furthermore, the design code specifies a maximum crack width of 0.2 mm for PSC members and 0.3 mm for reinforced concrete members. For the SC120f-PSC-NJ specimen, both the deflection and crack width at the center of the deck slab, generated at the serviceability limit state, satisfied the values specified in the design code (Figure 7). For the SC120f-PSC-J1 and J2 specimens, both the deflection and joint widening at the center of each specimen generated at the serviceability limit state also satisfied the specified limits (Figure 8).

## 4. Fatigue Performance Evaluation

### 4.1. Fatigue Loading Test and Residual Strength Test (SC120f-PSC-JF Specimens)

In the fatigue loading test (SC120f-PSC-JF), a total of two million fatigue load cycles were applied to the specimen using a 1000 kN dynamic actuator (MTS, Eden Prairie, MN, USA). When simple supports are applied as the boundary conditions for the fatigue loading test, a load imbalance can arise due to the impact of the load or the movement of the specimen during fatigue loading. Therefore, fatigue loads were applied after the specimen was fixed to the structural test frame with steel rods through the holes in the fixed part of the specimen (Figure 1 and Figure 9). In addition, to examine the effects of the accumulated fatigue, data were measured at the first cycle and then after 10^2^, 10^3^, 10^4^, 10^5^, 10^6^, and 2 × 10^6^ cycles. The fatigue loads were applied at a rate of 2–3 times/s using a fixed-point fatigue–loading method. The fatigue load (110 kN), obtained by multiplying the standard truck (KL-510) rear wheel load (96 kN) of the Korean Highway Bridge Design Code (Limit State Design) (2015) by the impact load factor of 15%, was applied to verify the performance in a more severe situation than the fatigue level required by design (80% of the standard truck rear wheel load was considered for fatigue examination). Moreover, the performance of the joint was verified in the harshest environment by applying the fatigue load to the joint interface of the specimen over a wheel–ground contact area of 231 mm × 577 mm. After 2 × 10^6^ fatigue–loading cycles, the residual strength test was conducted in the same manner as the static loading test to enable a comparison under the same conditions. Therefore, the boundary conditions were adjusted again in the same way as in the static loading test.

### 4.2. Result of Fatigue Loading Test and Residual Strength Test

#### 4.2.1. Failure Mode

As a result of applying fatigue loads to the SC120f-PSC-JF specimen, the joint slightly widened up to 2 × 10^6^ fatigue loading cycles, although no cracking was observed in the concrete deck slab body. In the residual strength test, a failure mode similar to those used for static loading tests (SC120f-PSC-J1 and J2) was observed, as shown in Figure 10.

#### 4.2.2. Evaluation of Flexural Strength

The load–deflection curve of the fatigue loading test specimen (SC120f-PSC-JF) obtained from the residual strength test is compared with those of the static loading test specimens (SC120f-PSC-J1 and J2) in Figure 11 and Figure 12. The maximum load applied to the SC120f-PSC-JF (304.53 kN) was similar to those of the static loading test specimens (322.07 and 289.04 kN). Moreover, the maximum load was approximately 3.47 times higher than the ultimate limit state load specified in the design code, indicating that the structure had a sufficient safety factor despite the accumulation of fatigue loads (Table 4).

### 4.3. Evaluation of Serviceability

The deflection of the SC120f-PSC-JF fatigue loading test specimen shown in Figure 13a, which shows the deflection at the serviceability limit according to the number of loads, was about 28.2–37.2% of the allowable deflection specified in the design code. Figure 13b shows that the joint widening of the specimen is 19.2–24.8% of the limit of 0.2 mm for a PSC member. These results indicate that the deck slabs tested in this study would exhibit no serviceability problems until 2 × 10^6^ fatigue cycles have occurred. Similarly, in the residual strength measurement test conducted after the occurrence of 2 × 10^6^ fatigue cycles, the deflection and cracking were within the limits specified in the design code (Figure 14).

## 5. Conclusions

This study verifies the structural performance of joints in thin long-span bridge deck slabs that used 120 MPa UHPC reinforced with steel fibers. The effect of the PSC–type–joint by post-tensioning methods on static and fatigue performance were verified experimentally. The following conclusions were derived:(1)In the static loading test, the flexural strengths of the specimens with joints were 74% and 66% of that of the specimen without a joint. However, they were about 3.7 and 3.3 times higher than the levels required by the design code.(2)In addition, the deflections and crack widths of the specimens with joints satisfied the limit of the design code at the serviceability limit state.(3)After two million fatigue load cycles were applied, the residual strength of the specimen with a joint was similar (95%, 105%) to the flexural strengths of the specimens with joints in the static loading test.(4)In the fatigue loading test, the deflection at the serviceability limit according to the number of loads, was about 28.2–37.2% of the allowable deflection specified in the design code, and the joint widening of the specimen was 19.2–24.8% of the limit.

## Figures and Tables

**Figure 1 materials-12-03040-f001:**
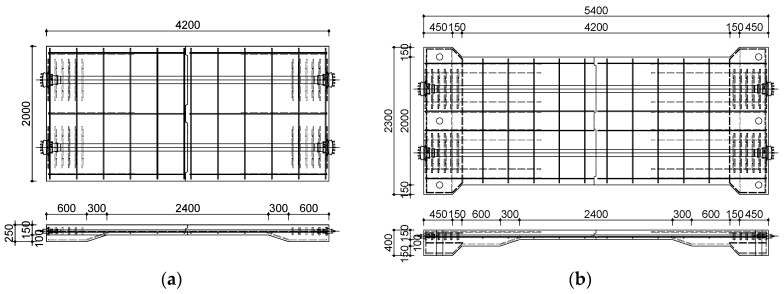
Dimensions of test specimens (unit: mm): (**a**) static specimens (SC120f-PSC-NJ, J1, J2); (**b**) fatigue specimens (SC120f-PSC-JF).

**Figure 2 materials-12-03040-f002:**
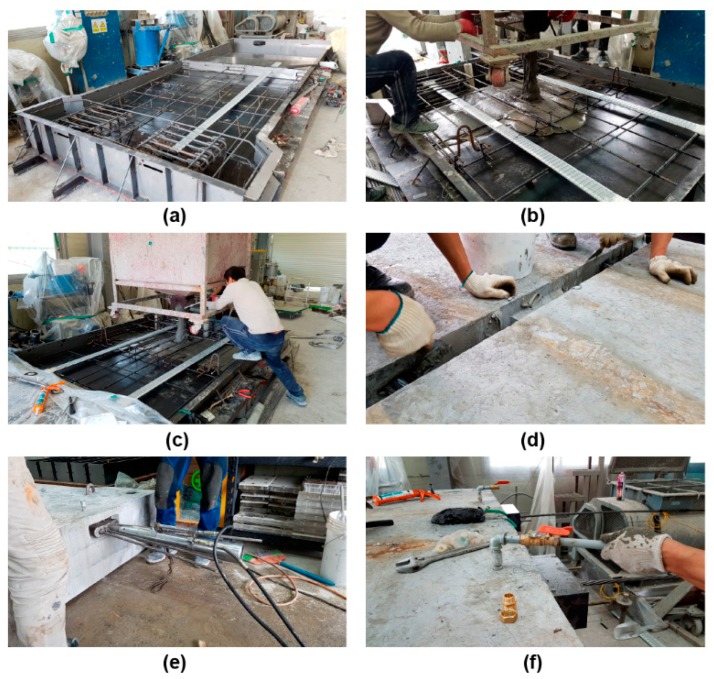
Fabrication of the specimens: (**a**) assembly of mold and reinforcement placing, (**b**) concrete casting (one side), (**c**) concrete casting (other side, match casting), (**d**) epoxy spreading, (**e**) jacking, and (**f**) grouting.

**Figure 3 materials-12-03040-f003:**
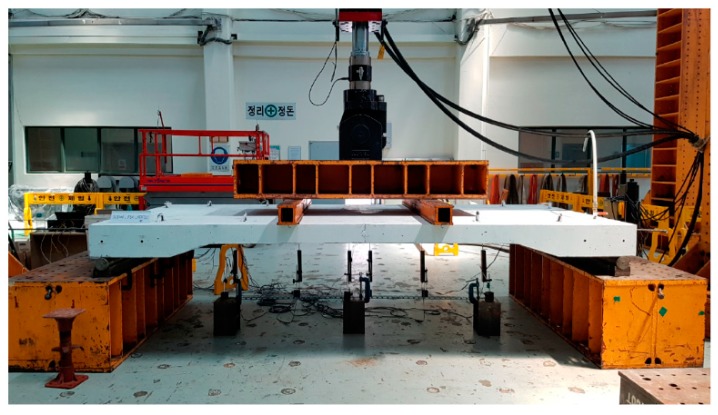
Static testing of specimens SC120f-PSC-J2.

**Figure 4 materials-12-03040-f004:**
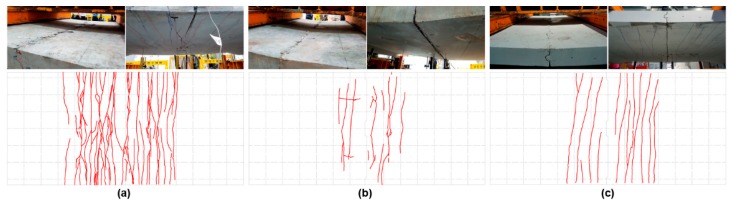
Photographs and maps of failure modes in static loading tests. (**a**) SC120f-PSC-NJ, (**b**) SC120f-PSC-J1, and (**c**) SC120f-PSC-J2.

**Figure 5 materials-12-03040-f005:**
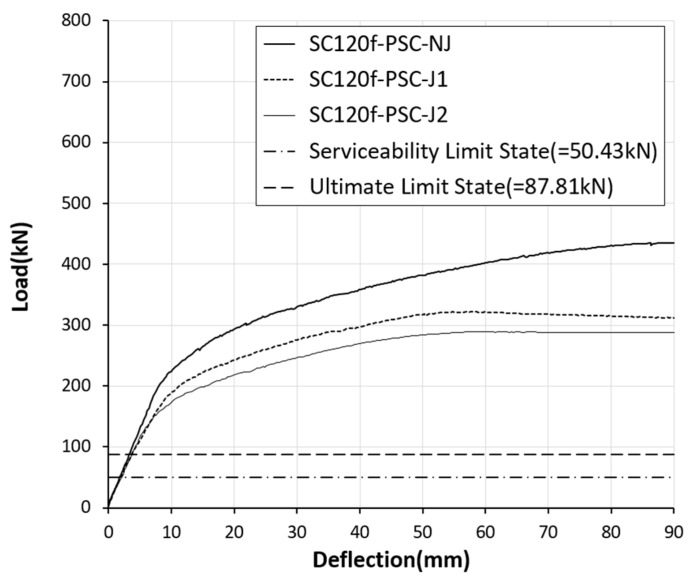
Load–deflection curves for static loading test specimens.

**Figure 6 materials-12-03040-f006:**
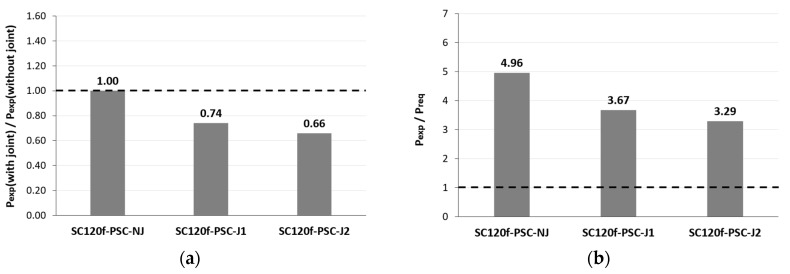
Comparison of maximum load for static loading test specimens; (**a**) P_exp_ (with joint)/P_exp_ (without joint), and (**b**) P_exp_/P_req_.

**Figure 7 materials-12-03040-f007:**
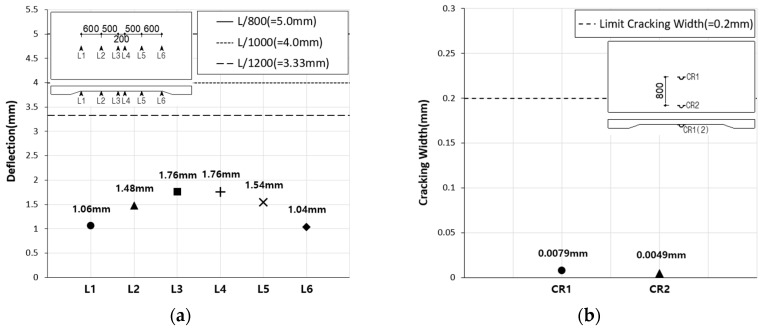
Deflection and crack width of SC120f-PSC-NJ (static loading test). (**a**) Deflection, and (**b**) crack width.

**Figure 8 materials-12-03040-f008:**
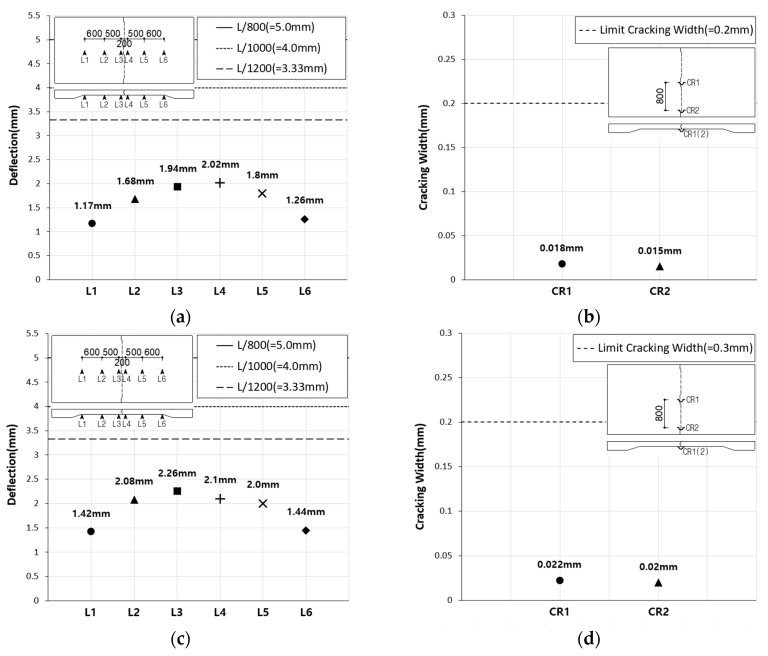
Deflection and crack width of SC120f-PSC-J1, J2 (static loading test). (**a**) Deflection (SC120f-PSC-J1), (**b**) crack width (SC120f-PSC-J1), (**c**) deflection (SC120f-PSC-J2), and (**d**) crack width (SC120f-PSC-J2).

**Figure 9 materials-12-03040-f009:**
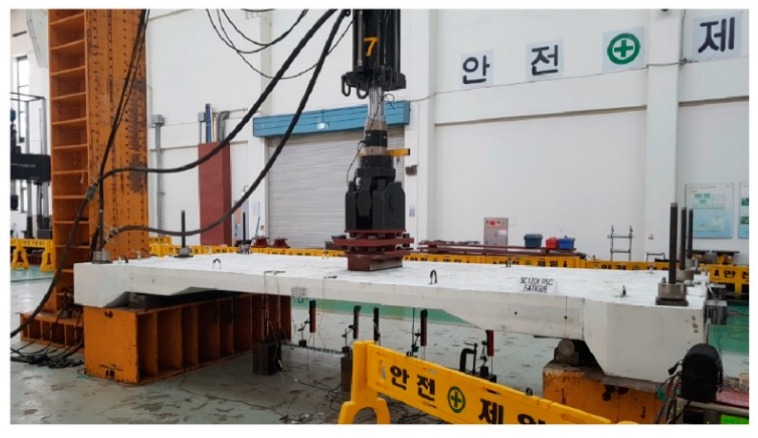
Fatigue testing of SC120f-PSC-JF specimen.

**Figure 10 materials-12-03040-f010:**
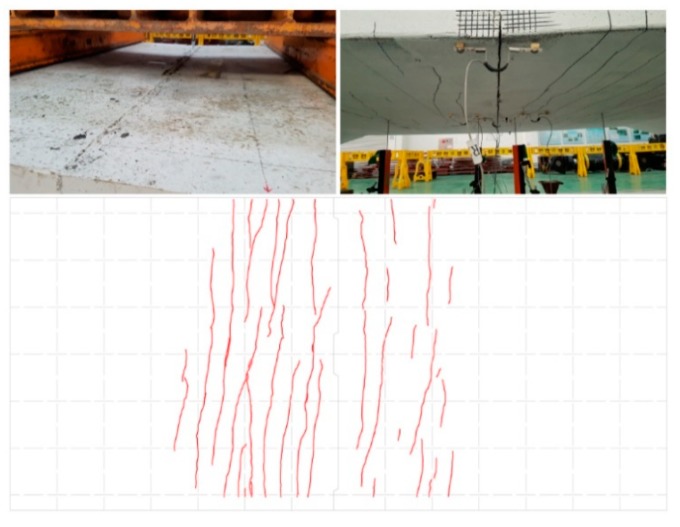
Failure mode of SC120f-PSC-JF.

**Figure 11 materials-12-03040-f011:**
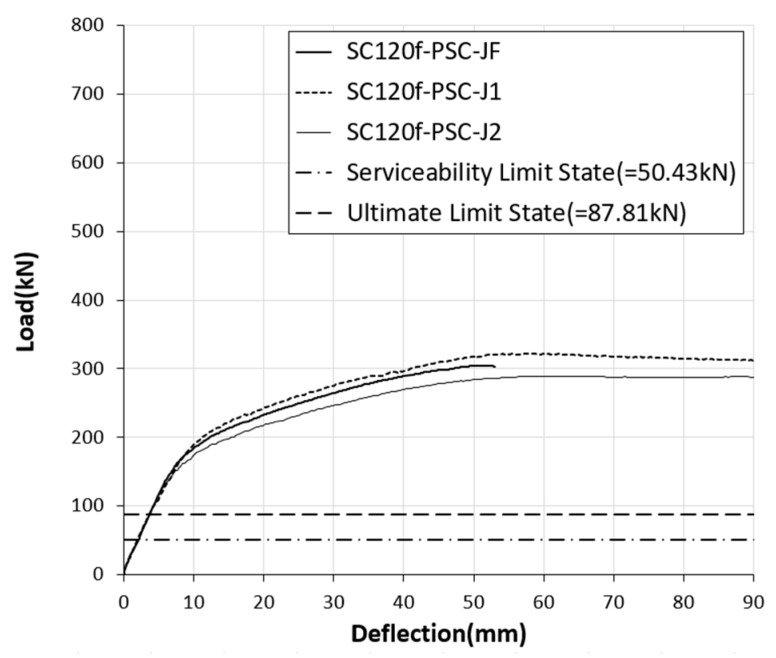
Load–deflection curves of test specimens with joints.

**Figure 12 materials-12-03040-f012:**
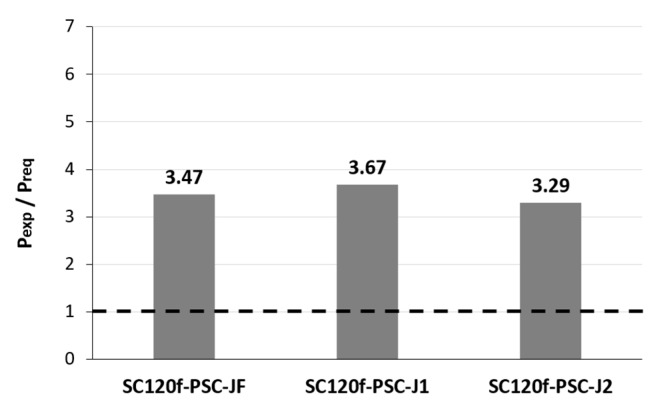
Comparison of maximum load for test specimens with joints.

**Figure 13 materials-12-03040-f013:**
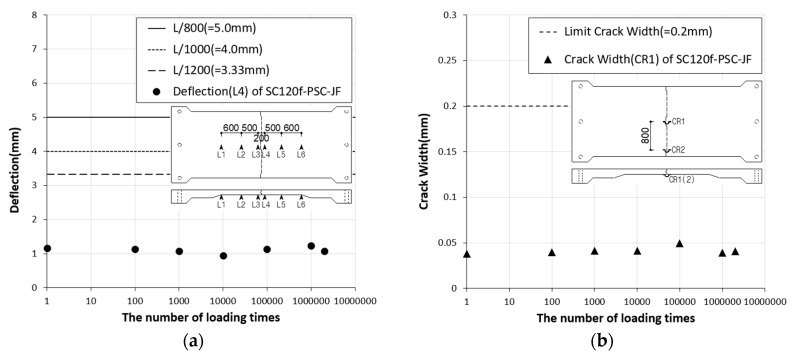
(**a**) Deflection and (**b**) crack width of SC120f-PSC-JF (fatigue test).

**Figure 14 materials-12-03040-f014:**
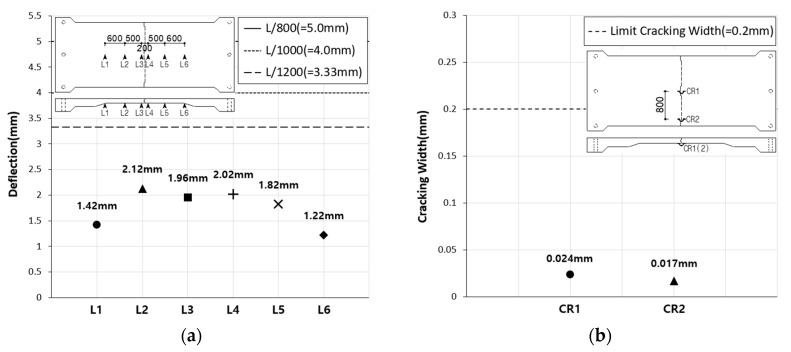
(**a**) Deflection and (**b**) crack width of SC120f-PSC-JF (residual strength test).

**Table 1 materials-12-03040-t001:** Summary of specimens used in this study.

Specimen	Joint	Test Type
SC120f-PSC-NJ	No	Static
SC120f-PSC-J1	Yes	Static
SC120f-PSC-J2	Yes	Static
SC120f-PSC-JF	Yes	Fatigue

**Table 2 materials-12-03040-t002:** Basic mix composition of SC120f.

W/B	Mass Unit (kg/m^3^)						
	Water	Premix Binder	Fine Aggregates	Steel Fiber		Super Plasticizer	Air-Entraining Agent
				16 mm	20 mm		
0.23	210	1180	847	-	78	7	17

Premix binder: cement, Zr, BS, filler, expansion agent, shrinkage reducing agent premix; Zr:BS = 30:70.

**Table 3 materials-12-03040-t003:** Comparison of maximum load for static loading test specimens.

Specimen	Deflection at Maximum Load	Maximum Load	Ultimate Limit State Load	P_exp_ (with Joint)/P_exp_ (without Joint)	P_exp_/P_req_
SC120f-PSC-NJ	88.06 mm	435.65 kN	87.81 kN	1.00	4.96
SC120f-PSC-J1	58.6 mm	322.07 kN		0.74	3.67
SC120f-PSC-J2	60.18 mm	289.04 kN		0.66	3.29

Note: P_exp_ (with joint) = experimental maximum load of SC120f-PSC-J1 or J2; P_exp_ (without joint) = experimental maximum load of SC120f-PSC-NJ; P_req_ = ultimate limit state load.

**Table 4 materials-12-03040-t004:** Comparison of maximum load for test specimens with joints.

Specimen	Deflection at Maximum Load	Maximum Load	Ultimate Limit State Load	P_exp_/P_req_
SC120f-PSC-JF	50.48 mm	304.53 kN	87.81 kN	3.47
SC120f-PSC-J1	58.6 mm	322.07 kN		3.67
SC120f-PSC-J2	60.18 mm	289.04 kN		3.29

Note: P_exp_ = experimental maximum load; P_req_ = ultimate limit state load.

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
