# Peer review of "Structural Safety Evaluation of Precast, Prestressed Concrete Deck Slabs Cast Using 120-MPa High-Performance Concrete with a Reinforced Joint"

_materials, 2019, doi:10.3390/ma12183040_

Round 1
Reviewer 1 Report
In this study, the authors verified the structural performance of prestressed concrete deck slabs prepared using 120-MPa high-strength fiber-reinforced concrete experimentally. They found that the deck slabs were found to satisfy all the structural performance. The authors also evaluated the structural performance of slabs with joints to verify the suitability of the deck slabs for use with the precast segmental construction method. This is a very interesting study and this study could be very useful for real applications. The only concern is that the focus of this study may be too narrow, so the authors may give some general introduction in the abstract and introduction parts so that more people could understand the importance of this study.
Reviewer 2 Report
The authors investigated the Structural Safety Evaluation of Precast, Prestressed Concrete Deck Slabs Formed Using 120-MPa High- Performance Concrete with a Reinforced Joint. The main conclusions were clearly proved by comparing the experimental results of the UHPC joints with those with no joints. Generally, this is interesting paper that addresses issues will be of interest to readers of the material Journal. However, important issues must be addressed before the final acceptance as follows:
The paper needs for proofreading since the language is very weak The introduction is very concise, and the reported up-to-date experimental results is not clear Analysis of the test results either under static load or fatigue loads is weak and more detailed discussion is needed The resolution of Figures1, and 7 is bad and need to be enhanced
5. The conclusions must be specific, and the current construction is not acceptable
Overall, the authors are advised to carefully read the following papers and follows the way of introducing the topic, reporting results, doing in-depth analysis and doing very specific and valuable conclusions
Gar, S., Head,, Hurlebaus, S. , and Mander, J. B. (2013) “Comparative Experimental Performance of Bridge Deck Slabs with AFRP and Steel Precast Panels.” Journal of Composites for Construction, 17(6), https://doi.org/10.1061/(ASCE)CC.1943-5614.0000380. Natário,F., Ruiz,F., Muttoni, A. (2015). “Experimental investigation on fatigue of concrete cantilever bridge deck slabs subjected to concentrated loads.” Engineering Structures, Volume 89, 191-203. Arafa, A., Farghaly, A., Ahmed, E., Benmokrane, B., (2016). “Laboratory Testing of GFRP-RC Panels with UHPFRC Joints of the Nipigon River Cable-Stayed Bridge in Northwest Ontario, Canada” Bridge Eng., 21(11), https://doi.org/10.1061/(ASCE)BE.1943-5592.0000943
Reviewer 3 Report
The authors presented an interesting issue related to the structural performance of a PSC deck slab fabricated using 120-MPa high-performance concrete reinforced with steel fibers. However the following comments should be solved before trying to publish the paper.
Abstract:
In abstract, answer the questions: What problem did you study and why is it important? What methods did you use? What were your main results? And what conclusions can you draw from your results? Please make your abstract with more specific and quantitative results while it suits broader audiences.
Introduction:
Introduction part is poor. There is not enough exemplary research related to the topic of the article. It is required to extend the literature review if the authors know other tests than those already contained in the work. The originality of the paper needs to be clarified. It is of importance to have sufficient results to justify the novelty of a high-quality journal paper.
Description of the specimens:
Was one deck slab belonging to the one kind of construction tested or more samples tested to obtain an average? The unit of dimensions shown in the figure 1 should be given. The concrete recipe must be provided. Was the concrete mix somehow compacted?
Static and Fatigue Performance Evaluation and :
Please indicate the manufacturer, city and country of each material and device (in brackets). This applies to the entire manuscript. Were the tests performed on the basis of a standard or on the basis of some publications? If so, please refer to them. It would be advisable to compare the obtained results with literature reports, because research review is poor. The smallest font that was used in Figures 6 and 7 (on the deck slab sketch) is hardly visible.
Conclusions:
Conclusions should be shortened. They should briefly present the most important observations, together with numerical and percentage values.
Literature:
The paper contains very few references. The overview of the research in the introduction and the discussion on the obtained own results should be broadened (compare them with the literature).
Reviewer 4 Report
This manuscript presents the structural safety evaluation of precast, prestressed concrete deck slab formed using 120-MPa high-performance concrete with a reinforced joint. However, the topic is interesting but the presentation of work is too weak. English should be completely modified and written in an understandable way. Many sentences are not understandable for readers. In addition, it is not clear to distinguish Experimental part from Result and Discussion section and everything is written together which make the work more difficult to follow. The interpretation of results is too superficial and confusing. I can give my overall recommendation after addressing all comments. There are also some minor comments:
-What is the difference between SC120f-PSC-J1 and SC120f-PSC-J2?
-Description of testing methods should be in the experimental section and should not present in this form.
-In section 3.3, which type of strength did you mean? compressive or flexural?
-Fig 5 and Table 2 present the same data. Please consider only one of them. In addition what are ①, ②, ③ in table 2?
-The same comment for Fig 10 and Table 3.
-With which program are the cracks (red lines) in Figure 9 plotted?
Round 2
Reviewer 2 Report
The manuscript was noticeably improved; the authors’ reply to the reviewer comments is also acceptable. However; the conclusion must be modified to be specific and concise for the reader. other minor comments is stated in the attached file

Author Response
Thank you very much for investing the time to second review our manuscript and for being encouraging. The conclusion modified to be specific and concise for the reader. Also, other minor comments has been revised. You can check the revisions made in the attached manuscript file.
Reviewer 3 Report
The authors responded to the reviewer's suggestions sufficiently, but I have another suggestion. In figures 6 and 12 the font is too small.
Author Response
Thank you very much for investing the time to second review our manuscript and for being encouraging. The font in fugures 6 and 12 modified. You can check the revisions made in the attached manuscript file.
Reviewer 4 Report
Authors could address my comments and questions and the manuscript can be sent to the further proceeding.
Author Response
Thank you very much for investing the time to second review our manuscript and for being encouraging.